# Neurotrophic Factors in Experimental Cerebral Acanthamoebiasis

**DOI:** 10.3390/ijms23094931

**Published:** 2022-04-29

**Authors:** Natalia Łanocha-Arendarczyk, Karolina Kot, Irena Baranowska-Bosiacka, Patrycja Kapczuk, Aleksandra Łanocha, Danuta Izabela Kosik-Bogacka

**Affiliations:** 1Department of Biology and Medical Parasitology, Pomeranian Medical University in Szczecin, 70-204 Szczecin, Poland; nlanocha@pum.edu.pl (N.Ł.-A.); kotkar@pum.edu.pl (K.K.); 2Department of Biochemistry and Medical Chemistry, Pomeranian Medical University in Szczecin, 70-204 Szczecin, Poland; ika@pum.edu.pl (I.B.-B.); patrycja.kapczuk@pum.edu.pl (P.K.); 3Department of Haematology and Transplantology, Pomeranian Medical University in Szczecin, 70-204 Szczecin, Poland; aleksandra.lanocha@pum.edu.pl; 4Independent Laboratory of Pharmaceutical Botany, Pomeranian Medical University in Szczecin, 70-204 Szczecin, Poland

**Keywords:** *Acanthamoeba* spp., brain-derived neurotrophic factor, nerve growth factor, neurotrophin-3, neurotrophin-4, cerebral cortex, hippocampus, immunological status

## Abstract

To date, no studies have addressed the role of neurotrophins (NTs) in *Acanthamoeba* spp. infections in the brain. Thus, to clarify the role of NTs in the cerebral cortex and hippocampus during experimental acanthamoebiasis in relation to the host immune status, the purpose of this study was to determine whether *Acanthamoeba* spp. may affect the concentration of brain-derived neurotrophic factor (BDNF), nerve growth factor (NGF), neurotrophin-3 (NT-3), and neurotrophin-4 (NT-4) in brain structures. Our results suggest that at the beginning of infection in immunocompetent hosts, BDNF and NT-3 may reflect an endogenous attempt at neuroprotection against *Acanthamoeba* spp. infection. We also observed a pro-inflammatory effect of NGF during acanthamoebiasis in immunosuppressed hosts. This may provide important information for understanding the development of cerebral acanthamoebiasis related to the immunological status of the host. However, the pathogenesis of brain acanthamoebiasis is still poorly understood and documented and, therefore, requires further research.

## 1. Introduction

Granulomatous amoebic encephalitis (GAE), caused by *Acanthamoeba* spp., is a rare opportunistic CNS infection for which mortality due to neurological complications exceeds 90% [1,2]. Over the last 30 years, researchers have reported 75 cases of patients with GAE caused by *Acanthamoeba* spp. [3]. Immunosuppression appears to be a predisposing factor for brain infections by *Acanthamoeba* spp. [1,4]. The mechanism by which *Acanthamoeba* traverses the blood–brain barrier (BBB) is not completely understood and may involve factors produced by the parasite (e.g., adhesins, proteases, phospholipases), as well as the host (e.g., interleukin beta (IL-β), tumor necrosis factor-alpha (TNF-α), gamma interferon (IFN-γ), host cell apoptosis) [1]. Our previous research has shown that *Acanthamoeba* spp. infection can change the levels of matrix metalloproteinases (MMP-2,-9) and the tissue inhibitor of MMPs (TIMP-1,-3) in the brain [4]. The increase in the activity of MMPs during acanthamoebiasis may be primarily the result of the inflammation process, probably from the increased activity of proteolytic processes, and, to a lesser extent, a defense mechanism preventing neurodegeneration. Disruption of cellular homeostasis within the brain tissue can be caused by exogenous factors such as parasites, leading to degradation of the brain structure and severe dysfunction, or even death. The first line of protection against pathogen invasions and neuronal damage are microglia cells [5,6]. Brain injury and/or immune stimulation leads to rapid activation of these cells, which shows an association with the pathogenesis of neurodegenerative diseases [7].

Neurotrophins (NTs) such as brain-derived neurotrophic factor (BDNF), nerve growth factor (NGF), neurotrophin-3 (NT-3), and neurotrophin-4 (NT-4) are indicated as biological agents with potential applications in the treatment of neurodegenerative disorders. They have the ability to maintain a normal neuronal structure and function and to stimulate neurite growth under physiological conditions, as well as when the nervous system is damaged [8,9].

In neuronal tissues, BDNF participates in neurogenesis, promoting neurite branching, stabilizing nerve endings, affecting glutamatergic and GABAergic neurotransmission, and has neuroprotective effects associated with its high-affinity binding to tropomyosin-receptor kinase B (TrkB) [10,11]. BDNF ensures the proper functioning of the hippocampus and neocortex. Some authors suggest that abnormal BDNF levels might be due to the chronic inflammatory state of the brain in certain disorders, as neuroinflammation is known to affect several BDNF-related signaling pathways [12]. Brain pathologies are usually associated with a downregulation of BDNF, resulting in reduced levels in the brain and blood [11,12]. BDNF is critical for neuron survival after injury. It is a neurotrophin that may be produced following inflammatory stimuli as a compensatory mechanism to minimize neuronal damage.

Higher concentrations of NGF have been seen in the hippocampus as a result of brain injury, in part mediated by the effects on astrocytes of pro-inflammatory mediators and cytokines produced by immune cells [7]. Following a brain injury, NGF levels significantly increase IL-8, which stimulates the secretion of this protein in astrocytic tissue cultures [8]. There is much less research on the role of NT-3 and NT-4 in the processes occurring in the brain. Neurotrophin-3 promotes the survival of neurons and the repair of nerves [13]. Yan et al. [14] have shown that NT-3 probably binds to BDNF to regulate neurogenesis and nerve survival. Neurotrophin-3 also potentiates the neurogenerative effects of NGF and BDNF following a CNS injury [15]. Neurotrophin-4 plays an important role in the development of the nervous system. NT-4 has a similar role to that of BDNF—it controls the survival and differentiation of vertebrate neurons [7,15].

The activity of neurotrophins has been observed in some protozoan infections. To date, neurotrophin concentrations have been examined in toxoplasmosis [16], cerebral malaria [17,18], leishmaniasis [19] and American trypanosomiasis [20]. Knowledge regarding the exact role of neurotrophins in the development of parasitic diseases is still scarce.

Understanding how neuroinflammation is involved in disorders of the brain, especially in disease onset and progression, can be crucial for the development of new strategies of treatment. To date, no studies have addressed the implications of the role of neurotrophins during cerebral acanthamoebiasis in immunocompetent or immunosuppressed hosts. Thus, to clarify the role of NT in the cerebral cortex and hippocampus during acanthamoebiasis in relation to the host immune status, the purpose of this study was to determine how brain acanthamoebiasis affects the concentrations of the neurotrophins BDNF, NGF, NT-3, and NT-4.

## 2. Results

### 2.1. BDNF in the Cerebral Cortex and Hippocampus during Acanthamoebiasis

We found significant upregulation of BDNF in the cerebral cortex of the *Acanthamoeba* spp.-infected immunocompetent mice (A) at 8 days post-infection (dpi), compared with the immunocompetent uninfected group (C) (U = 2; *p* = 0.03), while in the hippocampus, the BDNF level in the immunocompetent infected group (A) was higher at 8 dpi, lower at 16 dpi, and higher at 24 dpi (Figure 1) than the level in the uninfected group (C). In the immunosuppressed mice (AS), there was an upward trend in the levels of BDNF in the cerebral cortex and hippocampus in relation to the duration of the infection, but it was not statistically significant.

There were no significant differences in BDNF levels in the studied brain structures between immunocompetent and immunosuppressed *Acanthamoeba* spp.-infected mice.

### 2.2. Nerve Growth Factor (NGF) in the Cerebral Cortex and Hippocampus during Acanthamoebiasis

In the immunocompetent *Acanthamoeba* spp.-infected mice (A), there were significant differences in NGF levels at 8 dpi (U = 1, *p* = 0.02), 16 dpi (U = 2.0, *p* = 0.02), and 24 dpi (U = 0, *p* = 0.03) between the studied brain structures (Figure 2). At 8 dpi, we observed significant upregulation of NGF in the hippocampus of the *Acanthamoeba* spp.-infected immunosuppressed mice (AS), compared with the immunosuppressed uninfected group (CS) (U = 2, *p* = 0.02). We noted a significant upregulation of NGF level in the hippocampus of the *Acanthamoeba* spp.-infected immunocompetent mice (A), compared with the immunosuppressed, infected group (AS) at 8 dpi and (U = 0, *p* = 0.04) and 16 dpi (U = 5, *p* = 0.04).

### 2.3. NT-3 in the Cerebral Cortex and Hippocampus during Acanthamoebiasis

There was a statistically significant higher level of NT-3 in the cerebral cortex in the group of infected immunocompetent mice (A) at 8 dpi (U = 0, *p* = 0.05), compared with the uninfected immunocompetent mice (C) (Figure 3). While in the hippocampus of the immunosuppressed mice infected with *Acanthamoeba* spp. (AS), there was a tendency toward downregulation of NT-3, it was not statistically significant. There were significant differences in NT-3 levels between the cortex and hippocampus in the infected immunocompetent mice (A) at 16 dpi. (U = 4.0, *p* = 0.05) (Figure 3).

### 2.4. NT-4 in the Cerebral Cortex and Hippocampus during Acanthamoebiasis

The levels of NT-4 were similar in the cerebral cortex between the infected immunocompetent (A) and immunosuppressed (AS) mice. In the hippocampus, there was significantly higher NT-4 levels than in cerebral cortex in the infected immunosuppressed mice (AS) at 8 dpi (U = 0, *p* = 0.02) and 16 dpi (U = 4.0, *p* = 0.05). There was a downward trend in NT-4 level in relation to the duration of infection in the hippocampus in both the infected immunocompetent (A) and immunosuppressed (AS) mice, but the observed differences were not statistically significant. There were statistically significant differences in NT-4 levels between the cortex and hippocampus in the infected immunocompetent mice (A) at 16 dpi (U = 1.0, *p* = 0.02) (Figure 4).

Figure 5 presents the levels of neurotrophins as the acanthamoebiasis progresses.

## 3. Discussion

Neurotrophins play important roles in maintaining homeostasis in the CNS, where disturbances in their function can lead to a number of nervous system diseases [15,21]. The biological effects of neurotrophins depend on their concentrations and receptor affinities and can play diverse roles by interacting with other receptors and ion channels [7]. Many studies have shown that neurotrophins are key modulators of neuroinflammation, apoptosis, blood–brain barrier permeability, memory capacity, and neurite regeneration [15,22]. Similar to neurodegenerative diseases, parasitic infections such as amoebic encephalitis in the brain are characterized by multifactorial pathogenesis. Not much is known about the role and the concentrations of neurotrophins in brains infected by parasites such as free-living amoeba. This study demonstrated that BDNF and NT-3 play significant roles in the early stages of acanthamoebiasis in immunocompetent hosts. We also showed a significant involvement of nerve growth factor in acanthamoebiasis in immunosuppressed hosts. The level of NGF in the hippocampus was influenced by the host’s immunological status; higher levels of this NT were found in immunocompetent than in immunosuppressed hosts at 8 dpi and 16 dpi.

Moreover, this is the first study that shows that NT-3 and NGF show tissue specificity during cerebral acanthamoebiasis. Higher levels of these neurotrophins were observed in the hippocampus than those in the cerebral cortex in immunocompetent hosts, while NT-4 levels were greater in the hippocampus regardless of the immunological status of the host. We found no tissue specificity for BDNF—its levels were similar in the cortex and hippocampus.

Increased BDNF synthesis during the acute phase of meningitis could stimulate the proliferation of dentate granule cells and promote neurogenesis [22]. BDNF might be directly participating in the inflammatory response, and human immune cells could produce BDNF [23,24]. Some experimental animal models have shown that elevated concentrations of BDNF in the brain are responsible for some modifications of the host immune response to CNS viral infections [22,25].

In parasitic diseases, the role of neurotrophins is poorly known, but changes in BDNF availability could be involved in the pathogenesis of cerebral malaria [17,18]. It was found that a low level of BDNF may disrupt synaptic function and neural plasticity during infection, contributing to long-term cognitive and neurologic impairment. Moreover, Linares et al. [17] observed that as the severity of symptoms of cerebral malaria increased, BDNF mRNA progressively diminished in several brain regions, and this correlated with the symptoms. In this study, we observed significant upregulation of BDNF in the cerebral cortex at the beginning of acanthamoebiasis in immunocompetent mice (A). An increase in BDNF level in the brain may have neuroprotective effects (following its high affinity to binding to tropomyosin-receptor kinase B (TrkB), while a reduction may be related to a progressive process of atrophy and/or neuronal death, usually observed in prolonged cerebral acanthamoebiasis [26]. The neuroprotective effects of BDNF are mediated by activation of the TrkB/MAPK/ERK1/2/IP3K/ Akt pathway, which leads to attenuation of apoptosis and cell damage caused by oxidative stress [27]. Some authors noted that immune cells may induce neuroprotection by the production and local secretion of neurotrophic factors. CD4+ and CD8+ T lymphocytes, B lymphocytes, and monocytes in the human peripheral immune system can produce BDNF [28], which may be a compensatory mechanism in response to inflammation that induces apoptosis. With the progression of neurodegenerative diseases, the phenotype of lymphocytes changes, and they stop secreting BDNF; it is possible that this process occurs also in the late stages of acanthamoebiasis, which might explain the significant contribution of BDNF at the beginning of acanthamoebiasis. However, the clinical and biological mechanisms behind the lower BDNF levels in the late stages of acanthamoebiasis are not fully known.

NT-3 plays an important role in the development and normal functioning of the nervous system and is structurally linked to other neurotrophins such as BDNF and NGF [14]. NT-3 exerts functional effects by attaching to the TrkC receptor with high affinity. This neurotrophin has been implicated in a variety of neurodevelopmental processes including programmed cell death, neuronal differentiation, and the establishment of neuronal connections [15]. Hicks et al. [29] noted that a mild lateral fluid percussion brain injury in rats induced a decrease in NT-3 mRNA, demonstrating that even a mild traumatic brain injury (TBI) differentially alters neurotrophin levels in the hippocampus. In this study, a significantly higher level of NT-3 was shown in the cerebral cortex at 8 dpi in the group of infected immunocompetent mice (A), compared with those of the controls (C), which confirms the important role of NT-3 in the initial phase of acanthamoebiasis. In our previous study, we found that MMPs activity during cerebral acanthamoebiasis is significant to BBB integrity disorders and the migration of inflammatory cells and parasites [4]. It cannot be excluded that besides MMPs, other neurotrophic factors may regulate the survival of nervous tissue during *Acanthamoeba* spp. infection. Kuznievska et al. [30] found that the expression of BDNF and MMP-9 is modulated by synaptic activity, which is a critical signal activating important pathways for neuronal plasticity. BDNF and NT-3 are homologs. It has been noted also that NT-3 promotes MMP-9 expression [31]. In this study, we suggest that BDNF and NT-3 may have neuroprotective effects in the early phase of cerebral acanthamoebiasis for an immunocompetent host (A). Increased levels of these neurotrophins are most likely associated with increased MMP-9 activity at 8 dpi (which we found in our previous research on cerebral acanthamoebiasis, Appendix A) [4]. Some authors noted that proforms of neurotrophin are cleaved and activated by MMPs; MMP-9 converts pro-BDNF to mature BDNF, resulting in TrkB activation [5,32]. The likely role of BDNF and NT-3 in neuroprotection in cerebral acanthamoebiasis is also supported by the fact that immunocompetent mice with acanthamoebiasis at 8 dpi showed typical neurological symptoms such as circular marching and aggression [33]. Some studies suggest upregulation of NGF mRNA in astrocytes in models of traumatic injury, Parkinson’s disease (PD), and neuroinflammation [34,35]. Inflammatory cytokines (IL-1β, TNF-α, and IL-6) can induce the synthesis of NGF in neuronal and glial cells, as well as in epithelial, endothelial, connective, and muscle cells [15]. The upregulation of NGF can regulate innervation and neuronal activity of peripheral neurons, inducing the release of immune-active neuropeptides and neurotransmitters, and can directly influence innate and adaptive immune responses. NGF binds to both specific (TrkANGFR) and/or pan-neurotrophin (p75NTR) receptors to promote (autocrine/paracrine) downstream effects on the surrounding epithelial and also immune (mast cells, eosinophils, B/T cells, macrophages) cells. The normally low basal production of NGF is enormously upregulated during the inflammatory response, but how NGF and its receptors, TrkA and p75NTR, regulate cells and mediators during inflammatory responses is not yet well defined [7,36]. It has been observed that joint activation of p75NTR and Trk determines cell survival; nevertheless, in the absence of concomitant Trk stimulation, neurotrophins can more strongly induce programmed cell death through the p75NT receptor [7]. NGF has a variety of effects that can be either pro-inflammatory or anti-inflammatory [36]. During acanthamoebiasis, we observed probably the pro-inflammatory action of NGF, particularly in the immunocompromised hosts (AS). NGF is upregulated during the inflammatory response and lymphocytes have been shown to produce this NT [36]. In cerebral acanthamoebiasis, Fu et al. [37] showed a cell infiltrate that was 51% composed of T lymphocytes. Moreover, in immunocompromised patients with GAE, neurological changes were observed, and a histopathological examination showed a severe inflammatory process with multinucleated giant cells, histiocytes, and inflammatory cells including lymphocytes [38]. It is possible that the upregulation of NGF in an immunocompromised host (AS) in this study is related to T-cell activation during acanthamoebiasis. Similary to our results, Aloe and Fiore [39] in *Schistosoma mansoni* infected mice observed that brain granulomas are associated with a significant alteration in the constitutive expression of NGF. The authors suggested that the neuropathological dysfunctions in neuroschistosomiasis may be linked to changes in the NGF levels caused by local formation of granulomas.

Some authors suggest that an MMP/TIMP imbalance is implicated in the pathogenesis of CNS disorders involving inflammation [40]. Moreover, an imbalance in MMP/TIMP may lead to changes in NGF levels. In our previous study, in terms of the hippocampus of immunosuppressed hosts at the beginning of *Acanthamoeba* spp. infection, there was a statistically significant difference in the MMP-9/TIMP-1 ratio [4]. In this research, we found changes in NGF concentration at the same time of infection in immunosuppressed hosts. Noga et al. [41] noted that dexamethasone may down-regulate NGF levels. In this study, we used methylprednisolone to reduce the immunity of the host; both dexamethasone and methylprednisolone are corticosteroids. We observed a downregulating trend in NGF in immunosuppressed *Acanthamoeba*-infected hosts in the cerebral cortex and hippocampus. We observed significantly lower NGF levels in infected immunosuppressed mice (AS) at 8 dpi and 16 dpi, compared with infected immunocompetent hosts (A). Rosenberg et al. [42] reported that corticosteroids suppress the expression of MMP-9 in CSF during acute CNS inflammation. If MMP-9 activity is reduced by immunosuppressive drugs, it may also reduce the neuroprotective effects of neurotrophins. The role of NT-4 might be similar to BDNF because both interact with TrkB [43]. NT-4 has neuroprotective effects following cerebral ischemia and might play a role in long-term potentiation and plasticity [44,45]. Cordiero et al. [16] observed no significant differences in NT-3 and NT-4 levels between patients with ocular toxoplasmosis and the controls. Tokunaga et al. [46] found elevated NT-4 levels in only 36% of patients with bacterial encephalitis (4/11) and in 30% of patients with viral encephalitis. In this study, we also found no effect of cerebral acanthamoebiasis on NT-4 levels. However, it was noted that NT-4 showed tissue specificity during cerebral acanthamoebiasis, and the elevated levels occurred in the hippocampus regardless of host immune status. The occurrence of neurotrophins in specific brain structures is associated with the modification of their physiological functions [7]. Most commonly, neurotrophins have been detected in the hippocampus and cerebral cortex, indicating that these two areas are important targets of NT [15]. Some studies noted higher activity levels of NGF, NT-3, and NT-4 in the hippocampus, with BDNF the most abundant NT in the brain, mainly in the hippocampus and cortex [47], which was consistent with our experimental results.

This research has certain limitations. The data were collected based on animal models. In future studies, patient *post-mortem* samples of the *Acanthamoeba* spp. infected brains or the brain human cell line should be studied to improve the clinical significance of the presented results. The concentrations of NTs were examined only with ELISA kits; therefore, other methods, e.g., Western blot should be added to quantify the protein expression in the tissues and to confirm our results. Moreover, large dispersion of the NTs concentration in the cerebral cortex and hippocampus of mice infected with *Acanthamoeba* spp. presented in our study may have resulted from many confounding factors, such as individual differences in host susceptibility to *Acanthamoeba* spp. invasion, host–parasite interactions, the timing and duration of the immunosuppressive treatment, and the strain of the parasite.

## 4. Materials and Methods

### 4.1. Animal Model and Parasites

The experimental course of acanthamoebiasis has been described in detail in our earlier studies [4,48,49]. A similar model has been used by other researchers [26,50,51]. This study was carried out on 96 male BALB/c mice weighing approximately 23 g, from the Centre of Experimental Medicine, Medical University in Bialystok, Poland. The animal model was approved by the Local Ethics Committee for Experiments on Animals in Szczecin (No. 29/2015, dated 22 June 2015) and Poznań (No. 64/2016 dated 09 September 2016).

The mice were divided into 4 groups: (1) immunocompetent uninfected control group (C, *n* = 18); (2) immunocompetent *Acanthamoeba* spp.-infected mice (A, *n* = 30); (3) immunosuppressed *Acanthamoeba* spp.-infected mice (AS, *n* = 30); (4) immunosuppressed uninfected control group (CS, *n* = 18).

*Acanthamoeba* spp. strain AM 22 was isolated from bronchoaspiration of a patient with acute myeloid leukemia (AML) and acute septic shock. The patient presented atypical pneumonia, with a loss of weight and respiratory efficiency. In the radiological examination, interstitial changes were observed with a visible pulmonary swelling [52]. Strain AM 22 has pneumophilic properties [48] as well as neurophilic effects [4]. The AM22 strain was analyzed by molecular methods, and genotype T16 was detected [52]. The trophozoites were grown on agar plates (NN Agar) covered with a suspension of deactivated *Escherichia coli* according to standard parasitological methods [53]. Immunosuppression was performed by intraperitoneal administration of 0.22 mg (10 mg/kg) methylprednisolone sodium succinate (MPS, Solu-Medrol, Pfizer, Europe MA EEIG; cat. no.: W07908) in 0.1 mL of 0.9% saline at −4, −3, −2, −1 and 0 days before inoculation [50]. The mice in the infected groups (A and AS) were inoculated intranasally with 3 μL of a suspension containing 10–20 thousand amoebae. The control groups (C and CS) were given an equivalent volume of sterile solution (3 μL of 0.9% NaCl solution). Euthanasia of the *Acanthamoeba* spp.-infected mice was conducted at 3 time-points: 8, 16, and 24 days post-infection (dpi), depending on the clinical symptoms such as a lack of mobility, depression, aggression, turning in circles, tousled (matted) hair, anorexia, or emaciation (wasting) and the degree of infection, using a peritoneal overdose of sodium pentobarbital (Euthasol vet, FATRO Polska Sp.zoo, Kobierzyce, Poland) (2 mL/kg body weight) and subsequently necropsied. We reisolated numerous *Acanthamoeba* spp. trophozoites from the cerebral cortex and hippocampus, which confirms the invasion of amoebae into these tissues. The samples of the cerebral cortex and hippocampus were fixed in liquid nitrogen and then stored at −80 °C until biochemical analyses.

### 4.2. Sample Homogenization

The collected fragments of brain structures (hippocampus and cerebral cortex) were homogenized using a hammer in a liquid nitrogen medium. RIPA lysis buffer (pH 7.4 containing protease and phosphatase inhibitors) was used to further homogenize the tissues: 20 mM Tris base—0.79 g; 0.25 mM NaCl—0.8 g; 100 mM EDTA—1 mL; 10% NP-40—10 µL, 10% deoxycholic acid sodium salt—2.5 mL (cOmplete™, Mini Protease Inhibitor Cocktail, Roche, Switzerland, PhosSTOP™, Roche, Switzerland). The samples were then incubated for 2 h at 4 °C with constant shaking and centrifuged (20 min/15,000× *g*/4 °C). The supernatant was removed and stored at −80 °C for later analysis. The extracted material was later thawed at room temperature for the following analyses.

### 4.3. Protein Assay

The concentration of neurotrophins was calculated from the protein content of the samples as measured using a MicroBCAPierce™ kit (Thermo Fisher Scientific, Waltham, MA, USA) according to the manufacturer’s instructions, and a plate reader (BiochromAsys UVM 340) at 562 nm. The test kit is a high-precision set of reagents for determining the total protein concentration of a test sample, compared with a protein standard (albumin). It is a colorimetric method based on the formation of a violet-colored complex in an alkaline medium involving the reduction of Cu^2+^ to Cu^+^ using bicinchoninic acid (BCA).

### 4.4. Determination of BDNF Concentration

Brain-derived neurotrophic factor concentration was measured using a BDNF ELISA Kit (Wuhan Fine Biotech, Wuhan, China, Cat No. EM0020), according to the manufacturer’s instructions. Tissues were homogenized in phosphate-buffered saline (PBS) (0.01 M, pH = 7.4) and then centrifuged for 5 min at 5000× *g* to obtain the supernatant. The reaction mixtures were transferred to the ELISA microplate, following the manufacturer’s instructions. A sandwich enzyme-linked immune-sorbent mouse BDNF assay was designed to measure the concentration bound between a matched pair of antibodies. Samples, standards, and controls added to the appropriate wells bound the immobilized (capture) antibody. Then, a sandwich was created by adding a second antibody, along with a labeled substrate to induce a color change. The intensity of this signal was directly proportional to the concentration of BDNF present in the sample. BDNF activity was determined by measuring the absorbance at 450 nm using a BiochromAsys UVM 340 spectrophotometer. The concentration was expressed as pg/mg protein.

### 4.5. Determination of NGF Concentration

Nerve growth factor concentration was measured using the NGF ELISA Kit (Wuhan Fine Biotech, Wuhan, China, Cat No. EM0148), according to the manufacturer’s instructions. The tissues were homogenized in phosphate-buffered saline (PBS) (0.01 M, pH = 7.4) and then centrifuged for 5 min at 5000× *g* to obtain the supernatant. The principle of the method was analogous to that of the BDNF assay. NGF concentration was determined by measuring absorbance at 450 nm using a BiochromAsys UVM 340 spectrophotometer. The concentration was expressed as pg/mg protein.

### 4.6. Determination of NT-3 Concentration

Mouse neurotrophin 3 (NT-3) concentration was determined using a Mouse NT-3 (neurotrophin 3) (ELISA Kit (MyBioSource, San Diego, CA, USA), Cat No. MBS455618), according to the manufacturer’s instructions. Tissues were homogenized in ice-cold PBS (0.01 mol/L, pH 7.0–7.2). To obtain the supernatant, lysates were centrifuged for 5 min at 5000× *g*. The NT-3 assay plate had been coated with an antibody specific for NT-3 by the manufacturer. Standards and samples added to the wells reacted with the biotin-conjugated antibody. Subsequently, a color change was shown upon the addition of peroxidase (HRP) and TMB substrate. The reaction was terminated by adding sulfuric acid solution, and the color change was measured spectrophotometrically at 450 nm. The concentration of NT-3 in the samples was determined by comparison with a standard curve. The concentration was expressed as ng/mg protein.

### 4.7. Determination of NT-4 Concentration

Mouse neurotrophin 4 (NT-4) concentration was measured using a Mouse NT-4 (neurotrophin 4) (ELISA Kit (MyBioSource, San Diego, CA, USA), Cat No. MBS2505246), according to the manufacturer’s instructions. Tissues were homogenized in ice-cold PBS (0.01 M, pH = 7.4). To obtain the supernatant, lysates were centrifuged for 5 min at 5000× *g*. The methods were analogous to those of the NT-3 assay. The concentration was expressed as pg/mg protein.

### 4.8. Statistical Analysis

The obtained results were analyzed statistically using Statistica Software version 13.1 and Microsoft Excel 2019. Arithmetical means and standard deviations (SD) were calculated for each of the studied parameters. In order to assess differences between the parameters, Kruskal–Wallis ANOVA followed by Mann–Whitney-U tests were used. Differences were considered statistically significant at *p* < 0.05.

## 5. Conclusions

Some strengths of this paper can be outlined. This is the first study addressing neurotrophic factors according to the host immunological status in cerebral acanthamoebiasis. At the beginning of infection in immunocompetent hosts, BDNF and NT-3 may reflect endogenous attempts at neuroprotection against *Acanthamoeba* spp. infection. In immunosuppressed hosts, we noted a probable pro-inflammatory effect of NGF during acanthamoebiasis. We suspect that the signaling pathways important for cerebral acanthamoebiasis could interact with each other and depend on the host’s immunological status. However, the pathogenesis of cerebral acanthamoebiasis is still poorly understood and documented and, therefore, requires further research. This study may provide important information for understanding the development of GAE.

## Figures and Tables

**Figure 1 ijms-23-04931-f001:**
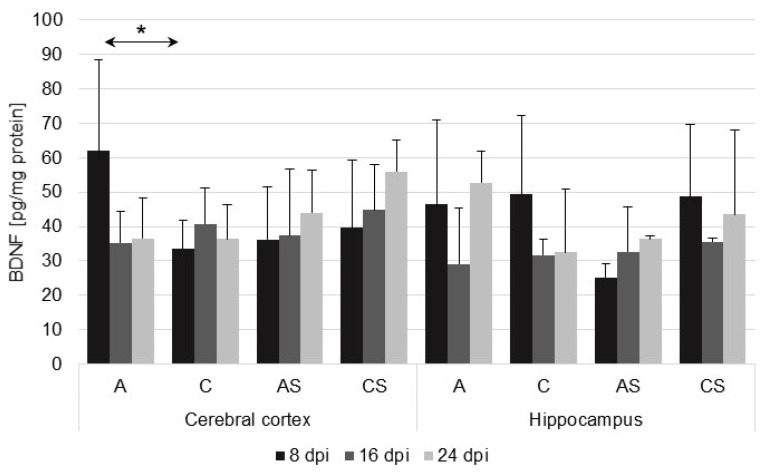
Brain-derived neurotrophic factor (BDNF) level (pg/mg protein) in the cerebral cortex and hippocampus in control and infected groups at 8, 16, and 24 days after *Acanthamoeba* spp. infection (dpi). Data present means ± SD for 6 independent experiments. C, immunocompetent uninfected mice; CS, immunosuppressed uninfected mice; A, immunocompetent *Acanthamoeba* spp. infected mice; AS, immunosuppressed *Acanthamoeba* spp. infected mice; solid arrows indicate differences between infected and control mice; * *p* ≤ 0.05 for the significance of difference (Mann–Whitney U test).

**Figure 2 ijms-23-04931-f002:**
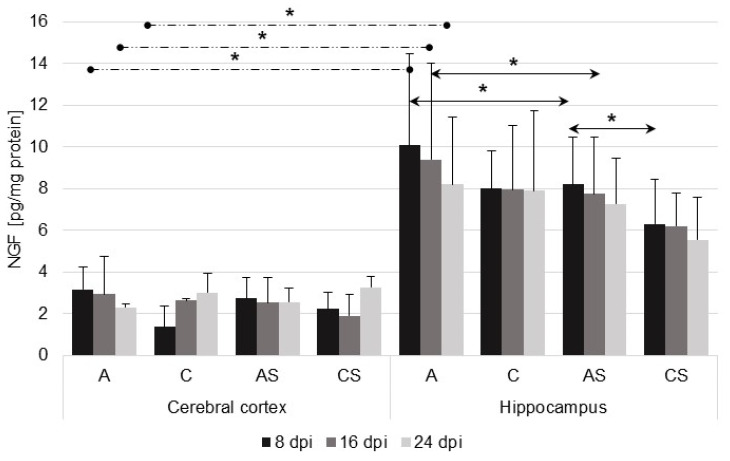
Nerve growth factor (NGF) level (pg/mg protein) in the cerebral cortex and hippocampus in control and infected groups at 8, 16, and 24 days after *Acanthamoeba* spp. infection (dpi). Data present means ± SD for 6 independent experiments. C, immunocompetent uninfected mice; CS, immunosuppressed uninfected mice; A, immunocompetent *Acanthamoeba* spp. infected mice; AS, immunosuppressed *Acanthamoeba* spp. infected mice; solid arrows indicate differences between infected and control mice while dashed arrows indicate differences between brain structures; * *p* ≤ 0.05 for the significance of difference (Mann–Whitney U test).

**Figure 3 ijms-23-04931-f003:**
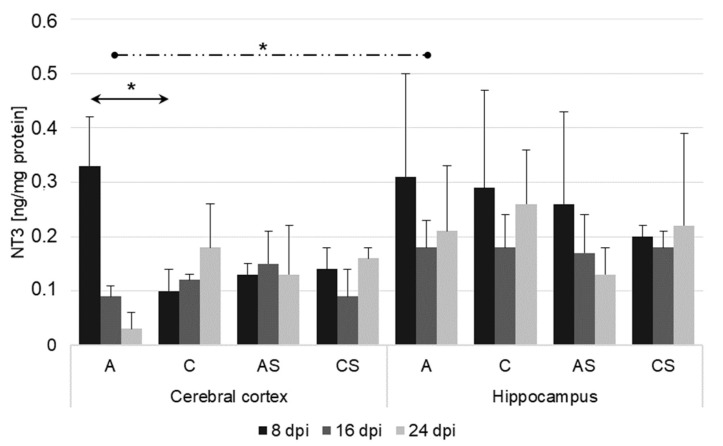
Neurotrophin-3 level (ng/mg protein) in the cerebral cortex and hippocampus in control and infected groups at 8, 16, and 24 days after *Acanthamoeba* spp. infection (dpi). Data present means ± SD for 6 independent experiments. C, immunocompetent uninfected mice; CS, immunosuppressed uninfected mice; A, immunocompetent *Acanthamoeba* spp. infected mice; AS, immunosuppressed *Acanthamoeba* spp. infected mice; solid arrows indicate differences between infected and control mice while dashed arrows indicate differences between brain structures; * *p* ≤ 0.05 for the significance of difference (Mann–Whitney U test).

**Figure 4 ijms-23-04931-f004:**
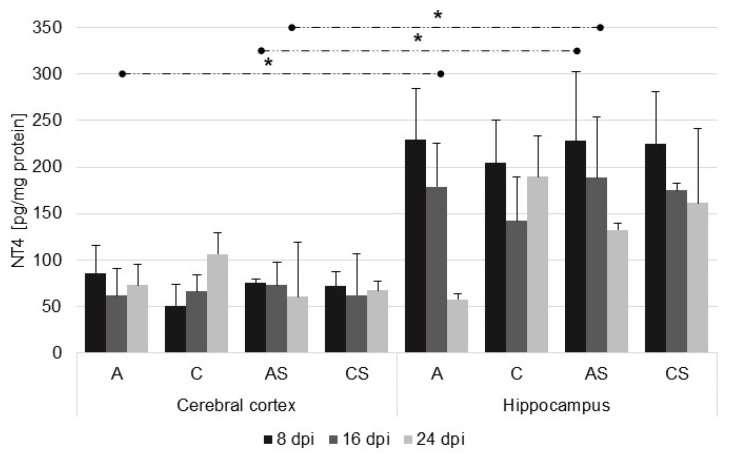
Neurotrophin-4 level (pg/mg protein) in the cerebral cortex and hippocampus in control and infected groups at 8, 16, and 24 days after *Acanthamoeba* spp. infection (dpi). Data present means ± SD for 6 independent experiments. C, immunocompetent uninfected mice; CS, immunosuppressed uninfected mice; A, immunocompetent *Acanthamoeba* spp. infected mice; AS, immunosuppressed *Acanthamoeba* spp. infected mice; dashed arrows indicate differences between brain structures; * *p* ≤ 0.05 for the significance of difference (Mann–Whitney U test).

**Figure 5 ijms-23-04931-f005:**
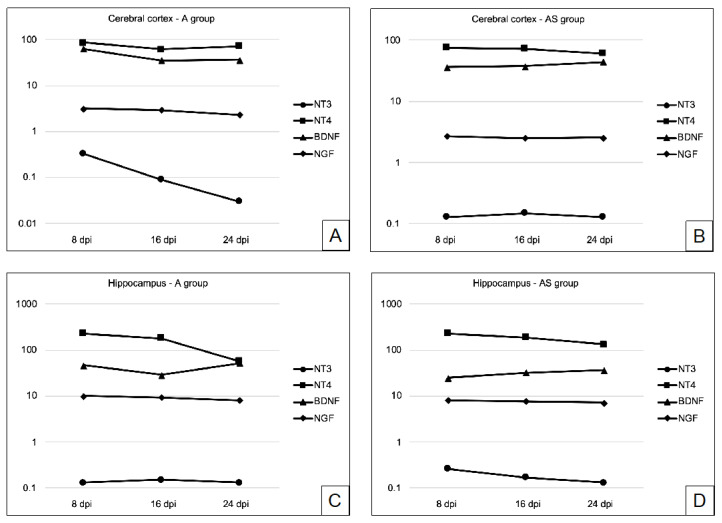
Concentrations of neurotrophin 3 (NT3, ng/mg protein), neurotrophin 4 (NT4, pg/mg protein), brain-derived neurotrophic factor (BDNF, pg/mg protein), and nerve growth factor (NGF, pg/mg protein) in the cerebral cortex of infected immunocompetent (A) and immunosuppressed mice (AS) (pictures (**A**) and (**B**), respectively) and in the hippocampus of infected immunocomptent (A) and immunosuppressed mice (AS) (pictures (**C**) and (**D**), respectively).

## Data Availability

Derived data supporting the finding of this study are available from the corresponding author (D.I.K.-B.) upon request.

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
