# Peer review of "Neurotrophic Factors in Experimental Cerebral Acanthamoebiasis"

_ijms, 2022, doi:10.3390/ijms23094931_

Round 1
Reviewer 1 Report
The present study aimed to evaluate the relationship between experimental cerebral acanthamoebiasis and the concentrations of neurotrophic factors (NTs) including BDNF, NGF, NT-3, and NT-4. The topic is interesting, however, several points limits the value and translational significance of this study.
- In this study, the authors established animal model in BALB/c mice, however, model validation data is not provided.
- The data were collected based on animal model, it is better to validate using patient samples to improve its clinical significance.
- The concentrations of NTs were examined using ELISA kit only, other methods eg. Western blot should be added to quantify the protein expression in tissues.
- This study is only a descriptive study with limited information to readers, no mechanistic findings were provided.
Author Response
Reviewer 1 (green color on manuscript)
Thank you for your review of our paper. We have answered each of your point below
The present study aimed to evaluate the relationship between experimental cerebral acanthamoebiasis and the concentrations of neurotrophic factors (NTs) including BDNF, NGF, NT-3, and NT-4. The topic is interesting, however, several points limits the value and translational significance of this study. Thank you
- In this study, the authors established animal model in BALB/c mice, however, model validation data is not provided. Thank you.
The diagnosis of Acanthamoeba spp. infection is poorly developed. It is mainly based on molecular studies or culture method, reisolation.The virulence of the amoebae was determined based on the degree of infection (on a scale 0-2) and determined as a ratio of the number of infected animals (presence of amoebae in studied tissues) to the number of inoculated mice, the table containing this results is given in our previously paper Kot et al. 2021 [Folia Biologica (Kraków) 2021; 69:4. Doi:10.3409/fb_69-4.18]. Moreover in Acanthamoeba infected mice we observed some neurological symptoms. We analysed activity of the mice, feeding, appearance of the fur, hunched position, ataxia and tremors. The table containing clinical signs found in all mice at 8, 16 and 24 days post-inoculation is given in the paper Kot et al. 2021 [Folia Biologica (Kraków) 2021; 69:4. Doi:10.3409/fb_69-4.18]. However, we added brief information in the material and method section. In our study, confirmation of Acanthamoeba spp. invasion into particular organ was checked by reisolating the amoebae from brain structures. We reisolated numerous Acanthamoeba spp. trophozoites from cerebral cortex and hippocampus, which confirms the invasion of amoebae into these tissues. We added this information in the main text. A similar disease model has also been used by other researchers, including Górnik and Kuźna-Grygiel [2005], Derda et al. [2006], and Wojtkowiak-Giera et al. [2016].
- The data were collected based on animal model, it is better to validate using patient samples to improve its clinical significance. Thank you.
The unknown mechanisms of brain Acanthamoeba invasion, the lack of an effective drugs and diagnostic methods necessitate intensive research on Acanthamoeba parasites on viable animal model. A similar disease model (with mice Balb/c) has also been used by other researchers, including Górnik and Kuźna-Grygiel [2005], Derda et al. [2016], and Wojtkowiak-Giera et al. [2016; 2018; 2019]. This model mimics several aspects of human brain infection caused by Acanthamoeba including neurological symptoms, macro- and microscopic changes in brain tissue [Górnik-Kuźna-Grygiel 2005; Khan 2010; Veríssimo Cde et al., 2013]. The route of infection to the CNS, incubation period, and histopathological lesions in animal model are analogous to those in humans (Khan 2006). Over the last 30 years researchers have reported 75 cases of patients with amoebic encephalitis caused by Acanthamoeba spp. To date, no case of brain infection caused by Acanthamoeba spp. has been reported in Poland. Due to the fact that it is a rare opportunistic invasion reported mainly in USA, obtaining biological samples from humans in Central Europe is basically impossible. Therefore, in our study we used a validated animal model.
- The concentrations of NTs were examined using ELISA kit only, other methods eg. Western blot should be added to quantify the protein expression in tissues. Thank you.
We agree with the Reviewer that more analysis would be more convincing but we performed biochemical (matrix metalloproteinases (MMP-2, MMP-9) and their tissue inhibitors (TIMP-1, TIMP-3), immunohistochemistry and molecular (LOX-5 and LOX-15 expression, results in press) analysis in the cerebral cortex and hippocampus of mice. Therefore, we had very little samples left. We decided to perform concentration of BDNF, NGF, NT-3, NT-4 in the brain structures. We have no more samples and we are not able to perform extra analysis. Doing new, additional analyses involves redoing the experiment and getting new approvals from the local ethics committee.
4. This study is only a descriptive study with limited information to readers, no mechanistic findings were provided. Thank you.
No studies to date have addressed the implications of the role of neurotrophins in Acanthamoeba spp. brain infections in immunocompetent or immunosuppressed hosts. The clinical and biological mechanisms behind the neurotrofins levels in acanthamoebiasis are still unknown Thus, to clarify the role of NT in the cerebral cortex and hippocampus during acanthamoebiasis in relation to the host immune status, the purpose of this study was to determine whether Acanthamoeba spp. may affect the concentrations of BDNF, NGF, NT-3, and NT-4 as biological agents with potential applications in the treatment of neurodegenerative disorders. We changed the discussion and we also added some information connected with possible NT mechanisms.
Reviewer 2 Report
Łanocha-Arendarczyk et al, explores the neurotrophins response in the brain in a mouse model of acanthamoebiasis. Those molecules can help prevent extensive damage to the brain and understanding their natural response is as such important for therapeutic intervention. The authors compare immunocompetent and immunosuppressed hosts and find that at least NGF has a lowered release in immunosuppressed infected animals.
The findings are novel, well written and an important for step for future therapeutic interventions against cerebral acanthamoebiasis. However, the way the data are presented and the high variablity make it difficult to know how sounds the findings are.
Given the very large number of data-points, it is really a shame that the figures are presented in box plot with error bars, individual data points or violin plots would help giving a better idea of the distribution giving the very large error bars. This could help clarify if there is an experiment repeat effect that explains the dispersion ? or another source of variability, maybe that could be discussed by the authors and corrected for if the source is known ( such as for experimental repetitions).
A multivariate analysis (MANOVA and/or principale composante analysis) considering all neurotrophins at once, might also help clarify the response pattern ( and would give more power to the analysis).
As a smaller comment, the authors discuss quite extensively the relationships between neurotrophins and MMPs, and as the authors report previous experience with MMPs, it would be very interesting to have a figure with both MMP and neurotrophins level (notably MMP-9 as it is seems important for the conversion of BDNF).
Author Response
Reviewer 2 (blue color on manuscript)
Thank you for your review of our paper. We have answered each of your point below
Łanocha-Arendarczyk et al, explores the neurotrophins response in the brain in a mouse model of acanthamoebiasis. Those molecules can help prevent extensive damage to the brain and understanding their natural response is as such important for therapeutic intervention. The authors compare immunocompetent and immunosuppressed hosts and find that at least NGF has a lowered release in immunosuppressed infected animals. Thank you.
The findings are novel, well written and an important for step for future therapeutic interventions against cerebral acanthamoebiasis. However, the way the data are presented and the high variablity make it difficult to know how sounds the findings are Thank you.
Given the very large number of data-points, it is really a shame that the figures are presented in box plot with error bars, individual data points or violin plots would help giving a better idea of the distribution giving the very large error bars. Thank you.
Acanthamoebiasis animal model was performed on 96 male mice, however in the present study, we used only 6 mice from each group. The cerebral cortex and hippocampus are small structures of the mouse brain and we wanted to perform as many analyzes as possible. Apart from neurotrophin concentrations, we also analyzed MMPs and TIMPs analyses (ELISA, Western-blot, and IHC methods) and expression of LOXs (by PCR method). Due to the fact that we use only 6 mice (the amount of mice were calculated in statistical software: www.biomath.info), we think that individual data points or violin plots wouldn’t show the results better and we decided to keep the figures in this form.
This could help clarify if there is an experiment repeat effect that explains the dispersion ? or another source of variability, maybe that could be discussed by the authors and corrected for if the source is known ( such as for experimental repetitions).
We agree with the Reviewer that data distribution give the very large dispersion and it’s a limitation of a study. The pathological process occurring within the brain as a result of the infection by Acanthamoeba spp. is not fully understood. Many studies on acanthamoebiasis in experimental animals have reported individual differences in susceptibility to invasion, hence a possible source of data variability (Martinez and Kasprzak 1980). Even in experimental conditions, parasite models involve several often uncontrolled parameters, including the relationship between route of administration, timing and duration of immunossupresive treatment, mouse lineage, host–parasite interactions, and type of immune response (Chatelain and Konar 2015). Moreover the use of a parasite strain that induces brain lesions in a experimental GAE model is advisable. We used strain isolated from the bronchoaspirate of a patient with acute myeloid leukemia (AML) which has pneumophilic properties. Standardization of the parasite strain used in the study could help reduce variability in experimental outcomes. We added this information in the main text.
A multivariate analysis (M ANOVA and/or principale composante analysis) considering all neurotrophins at once, might also help clarify the response pattern ( and would give more power to the analysis). We added a figure in which we presented the response patters of all neurotrophins at different points during acanthamoebiasis.
Fig. 1. Concentration of neurotrophin 3 (NT3), neurotrophin 4 (NT4), brain-derived neutrophic factor (BDNF) and nerve growth factor (NGF) in the cerebral cortex of immunocompetent (A) and immunosuppressed mice (B) infected with Acanthamoeba
As a smaller comment, the authors discuss quite extensively the relationships between neurotrophins and MMPs, and as the authors report previous experience with MMPs, it would be very interesting to have a figure with both MMP and neurotrophins level (notably MMP-9 as it is seems important for the conversion of BDNF). We added figure concerning MMP-9, BDNF, NT-3 in the cerebral cortex and hippocampus of immunocompetent and immunosuppressed Acanthamoeba spp.-infected mice. The figure is presented in the supplementary material.
Fig. S1. Concentration of neurotrophin 3 (NT3), brain-derived neutrophic factor (BDNF) [present study] and matrix metalloproteinase 9 (MMP-9) [Łanocha-Arendarczyk et al. 2018] in the cerebral cortex of immunocompetent (A) and immunosuppressed mice (B) infected with Acanthamoeba spp. as well as in the hippocampus of immunocompetent (C) and immunosuppressed mice (D) infected with Acanthamoeba spp. in different days post infection (dpi)
Round 2
Reviewer 1 Report
The explanations should be incorporated in the discussion/limitation section.
Author Response
Reviewer 1 (red color on manuscript)
Thank you for your review of our paper. We have answered each of your point below
The explanations should be incorporated in the discussion/limitation section.
We added the information.